# Diagnosis-Specific Work Disability before and after Lumbar Spine Decompression Surgery—A Register Study from Sweden

**DOI:** 10.3390/ijerph18178937

**Published:** 2021-08-25

**Authors:** Thomas E. Dorner, Ellenor Mittendorfer-Rutz, Magnus Helgesson, Tea Lallukka, Jenni Ervasti, Konstantinos Pazarlis, Annina Ropponen, Pia Svedberg, Mo Wang, Syed Rahman

**Affiliations:** 1Division of Insurance Medicine, Department of Clinical Neuroscience, Karolinska Institutet, SE-171 77 Stockholm, Sweden; thomas.dorner@meduniwien.ac.at (T.E.D.); ellenor.mittendorfer-rutz@ki.se (E.M.-R.); magnus.helgesson@ki.se (M.H.); tea.lallukka@helsinki.fi (T.L.); annina.ropponen@ttl.fi (A.R.); Pia.Svedberg@ki.se (P.S.); mo.wang@ki.se (M.W.); 2Department of Social and Preventive Medicine, Centre for Public Health, Medical University of Vienna, Kinderspitalgasse 15/I, 1090 Wien, Austria; 3Department of Public Health, University of Helsinki, P.O. Box 20, 00014 Helsinki, Finland; 4Finnish Institute of Occupational Health, P.O. Box 18, FI-00032 Työterveyslaitos, Helsinki, Finland; jenni.ervasti@ttl.fi; 5Department of Surgical Sciences, Division of Orthopaedics, Uppsala University Hospital, Akademiska Sjukhuset Ing 70, SE-751 85 Uppsala, Sweden; konstantinos.pazarlis@surgsci.uu.se; 6Stockholm Spine Center, 194 89 Upplands Väsby, Sweden

**Keywords:** sick leave, disability pension, spine surgery, dorsopathy, low back pain

## Abstract

Low back pain (LBP) patients undergoing lumbar spine decompression surgery (LSDS) often suffer from multi-comorbidity and experience high work disability. This study aimed to identify diagnosis-specific work disability patterns in all LBP-patients before and after LSDS during 2008–2010, that were aged 19–60 years and living in Sweden (*n* = 10,800) and compare these patterns to LBP-patients without LSDS (*n* = 109,179), and to matched individuals without LBP (*n* = 472,191). Work disability days (long-term sickness absence (LTSA), disability pension (DP)) during the three years before to three years after the cohort’s entry date were identified by generalised estimating equations. LBP-patients undergoing LSDS had higher overall work disability during the three years following surgery (LTSA: 23.6%, DP: 6.3%) than LBP-patients without LSDS (LTSA: 19.5%, DP: 5.9%), and those without LBP (LTSA: 7.9%, DP: 1.7%). Among patients undergoing LSDS, the prevalence of work disability due to dorsopathies increased from 20 days three years before surgery to 70 days in the year after and attenuated to 30 days in the third year following surgery. Work disability for other diagnoses remained stable at a low level in this group (<10 days annually). LBP-patients undergoing LSDS have an unfavourable long-term work disability prognosis, primarily due to dorsopathies. Decompression surgery seemed to restrict further inclines in work disability in the long run.

## 1. Introduction

Low back pain (LBP) is a leading diagnosis for work disability (i.e., sickness absence (SA) and disability pension (DP)) [1,2,3,4]. Previous research revealed that LBP-patients undergoing spine surgery, such as lumbar spine decompression surgery (LSDS), have an even worse prognosis in terms of work disability compared to LBP-patients without such surgery [5]. Approximately 2% of Swedish residents undergo lumbar spine surgery during their lifetime [4]. When, and whether or not, patients with LBP with certain disc disorders should receive surgical intervention or undergo conservative treatment options is, however, an issue of ongoing debate [6,7]. Examining clinical characteristics as endpoints, especially with a short-term follow-up, has shown inconclusive results or trends towards a slightly better prognosis for patients who are treated by surgical procedures compared to those treated conservatively [8,9]. Longitudinal studies, examining LBP-patients who underwent LSDS vs. those who did not undergo LSDS, on the outcome of work disability, are scarce. However, studies such as these may contribute considerably to the current knowledge base and, eventually, towards decision making regarding methods of treatment for LBP. About 15% of LBP-patients treated with LSDS, who had a high work disability level before surgery, showed improvement regarding work disability already in the year after LSDS [5]. The same study revealed that a younger age, higher education, low pain intensity, and lack of mental co-morbidities were factors associated with a better work disability prognosis following decompression surgery [5].

Many patients suffering from lumbar spine disorders have various co-morbidities, such as common mental disorders (CMDs) [5], and in many cases it is the interaction between the spine disorder and the co-morbidity that leads to adverse outcomes, like work disability [10,11,12]. Additionally, not necessarily all individuals with LBP, with or without LSDS, who experience work disability will have SA or DP due to musculoskeletal diseases only. This is because the diagnoses for work disability in individuals initially diagnosed with back pain tend to change over time [13]. Therefore, to acquire a better picture of the actual disability burden following a spine surgery, it is important to consider SA and DP not only due to musculoskeletal diseases, but also for various other diagnoses. 

Socio-demographic factors [5,14], as well as different clinical characteristics are associated with a risk for both LSDS and work disability [5,10]. Therefore, it is crucial to consider them when examining work disability in patients undergoing LSDS. Such socio-demographic factors comprise sex, age, education level, living area, region of birth, and family situation, and clinical factors include somatic and mental co-morbidities, previous sickness absences, and dosages of analgesic and other drugs. Some of the aforementioned clinical factors can also be considered as proxy for the severity of the underlying disease. 

The aims of the study were to investigate the pattern of diagnosis-specific work disability in patients before and after lumber spine decompression surgery, adjusting for socio-demographic and clinical factors, and to compare these patterns across LBP-patients with LSDS, LBP-patients without LSDS, and individuals without diagnosed LBP. 

## 2. Materials and Methods

### 2.1. Registers

This study utilises nationwide data from five registers which are linked based on the personal identity numbers of all residents in Sweden [15] and provided by the following three different Swedish agencies: (1) Statistics Sweden, which provided data on age, sex, educational level, family situation, living area, country of birth and emigration, extracted from the longitudinal integrated database for health insurance and labour market studies (LISA) [16]; (2) The National board of Health and Welfare which provided data on the date and type of surgery based on the Swedish version of NOMESCO codes of surgical procedures, and International Classification of Diseases version 10 (ICD-10) diagnostic codes and dates for hospitalisation and specialised outpatient care, obtained from the National Patient Register [17,18], medication purchases based on the Anatomic Therapeutic Chemical (ATC) classification system with defined daily doses (DDD) taken from the Prescribed Drug Register [19], and date and cause of death recorded in ICD-10 codes, acquired from the Cause of Death Register [20]; (3) Swedish Social Insurance Agency which provided data on the dates and ICD-10 codes for underlying diagnoses for SA and DP, and duration of SA, obtained from the Micro-data for Analysis of Social Insurance (MiDAS) register.

### 2.2. Study Populations

All individuals registered in Sweden on 31 December preceding the year of entering the cohort, aged 19–60, with a diagnosis of LBP from in- or specialised outpatient care during 2008–2010 were included. LBP was defined by the following ICD-10 codes [21]: *Spondylopathies*: M47.8, M47.9, M47.9K, M48.0, M48.0K, M48.8, M48.8K and M48.8W; *Radiculopathies*: M51.1, M51.1K, M54.1; and *other diagnosis with LBP*: M53.8, M53.9, M54.3, M54.4, and M54.5. The LBP-patients were then stratified according to the type of treatment provided: by LSDS (*n* = 10,907) (NOMESCO surgical codes: ABC 7, 16, 26, 36, 56, 66, 99), and treated conservatively (*n* = 111,473). Each patient with LBP was matched (exact matching without any replacement) on age, sex, educational level and region of birth with up to four reference individuals, randomly selected from the general population, who had not been treated at in- or specialised outpatient care during 2001–2013 due to any degenerative spine disease (ICD 10: M40-M54) as the main or secondary diagnosis (*n* = 489,124). The cohort entry date (CED) for LBP-patients treated by LSDS was considered to be the date of surgery, for LBP-patients not treated by LSDS it was the date of diagnosis. Individuals with no diagnosed LBP were included on the identical CED to that of their matched LBP-patients. Those individuals who died or emigrated during the study period (from three years before inclusion to three years after) were excluded, leaving us with 10,800 LBP-patients with LSDS, 109,179 LBP-patients without LSDS and 472,191 individuals with no diagnosed LBP. 

### 2.3. Outcome Measures

Work disability was measured as the mean of the annual sum of net days with SA or DP due to a specific diagnosis. Net days considers the grade of SA and DP. As such, if an individual had, for example, 50% SA or DP for two days, this was counted as one net day. Hereafter, ‘work disability net days’ will be regarded as ‘work disability days’. The Swedish Social Insurance Agency keeps records of SA cases with a duration of 14 or more gross days for employed individuals, therefore, our study included SA spells lasting for 14 or more days. SA and DP were measured for three years before, and up to three years after, CED. The underlying main diagnosis for the SA spell or DP was determined by the ICD-10 codes. Diagnosis-specific annual work disability days were categorised as: due to CMDs (ICD 10: F32-33, F40-43), due to other mental diagnoses than CMDs (ICD 10: F00-99, except F32-33, F40-43), due to degenerative spine diseases (ICD-10: M40-54), due to musculoskeletal diseases other than degenerative spine diseases (ICD 10: M00-99 except M40-54), and due to other somatic diagnoses (ICD 10: all somatic diagnoses except musculoskeletal diseases, uncomplicated delivery (O80), external causes of morbidity (V01-X59), factors influencing health status and contact with health services (Z00-Z99) and codes for special purposes (U00-U85)). 

Additionally, we estimated that the most common reasons for work disability during the three years following CED, measured as first long-term SA (LTSA) spell (more than 90 net days) and DP, were due to: neoplasms (C00-D48), mental disorders (F00-F99), diseases of the nervous system (G00-G99), cardiovascular diseases (I00-I99), musculoskeletal diseases (M00-M99), symptoms, signs and abnormal clinical laboratory findings (R00-R99), injuries/poisoning (S00-T98), and all others. 

### 2.4. Covariates 

Socio-demographic factors such as sex, age, educational level, living area, region of birth, and family situation were measured on 31 December in the year preceding inclusion. The socio-demographic factors were categorised as mentioned in Table 1. 

As clinical covariates, previous in- or specialised outpatient care due to different diagnoses, SA in the year prior to inclusion, and the usage of analgesics, anxiolytics, antidepressants, and sedatives/hypnotics in the year prior to inclusion were used. Previous in- or specialised outpatient care was categorised as common mental disorders (ICD-10: F32-33, F40-43), other mental disorders (ICD-10: F00-99, except for F32-33, F40-43), LBP (ICD-10: M47.8, M47.9, M47.9K, M48.0, M48.0K, M48.8, M48.8K, M48.8W, M51.1, M51.1K, M53.8, M53.9, M54.1, M54.3, M54.4 and M54.5), degenerative spine diseases other than LBP (ICD-10: M40-54 except codes for LBP, as mentioned above), and somatic diagnoses other than musculoskeletal diseases (ICD 10: all somatic diagnoses except musculoskeletal diseases, uncomplicated delivery (O80), external causes of morbidity (V01-X59), factors influencing health status and contact with health services (Z00-Z99) and codes for special purposes (U00-U85)). Sickness absence in the year prior to inclusion was categorised into four categories: 1–90 days, 91–180 days, 181–365 days, and more than 365 days (i.e., SA spell initiated more than one year prior to inclusion). Prescribed and dispensed medication for analgesics, anxiolytics, antidepressants, and sedatives/hypnotics in the year prior to inclusion were coded according to the ATC codes: N02A and N02B; N05B, N06A, and N05C, respectively [22]. These were categorised, based on the annual prescribed and dispensed daily doses (DDD) of medication, into: low dose (>0–0.5 DDD), medium dose (>0.5–1.5 DDD) and high dose (>1.5 DDD). 

### 2.5. Statistical Analyses

The differences in clinical factors between the LBP-patients with and without LSDS, and individuals without LBP diagnosis were assessed using the Chi-square test. To adjust for socio-demographic variations between the compared groups, the absolute levels of annual diagnosis-specific mean work disability days were calculated by the least square means over a six-year period. The three years of observation for both before and after the t0 (cohort entry date) comprised t − 3 to t − 1 and t + 1 to t + 3, respectively. To estimate the slope and risk of diagnosis-specific work disability days before (t − 3 to t − 1), around (t − 1 to t + 1) and after (t + 1 to t + 3) the CED, repeated measure regression with a generalised estimating equations (GEE) method with an exchangeable correlation structure was applied in order to produce relative risks (RRs) and 95% confidence intervals (CIs) [23]. The estimates were calculated based on the differences in the adjusted means between the compared time points within a specific group (i.e., ‘LSDS’, ‘LBP no LSDS’, ‘No diagnosed LBP’). A slope was considered to be upward if the RR was >1 and downward if RR was <1. The differences in slopes between the different groups were assessed by their respective CIs. The slopes were considered similar if CIs overlapped. All regression analyses were controlled for all the mentioned socio-demographic variables and were conducted in SAS v.24.

## 3. Results

In Table 1 the socio-demographic characteristics of patients undergoing LSDS are shown. There were slightly more women (53%) than men the number of patients increased with age group, mostly patients had a medium level of education (52%), patients were mainly living in big (38%) or medium-sized cities (34%), the majority of patients were Swedish born (85%), and patients were commonly single without children living at home (37%) or married with children living at home (31%). The clinical parameters of LSDS patients and the two comparison groups are shown in Table 2. LBP-patients without LSDS were the group with the highest use of health care due to mental diagnoses. However, a significantly higher proportion of patients with LSDS were previously treated for LBP and other degenerative spine diseases, had the most SA days prior to inclusion, and received the highest dosages of analgesics, anxiolytics, antidepressants, and sedatives/hypnotics the year prior to inclusion than the LBP-patients without LSDS or individuals without diagnosed LBP.

Adjusted mean work disability days due to different diagnoses three years prior to three years after inclusion in the three different groups are shown in Figure 1. Patients with LSDS mainly had work disability days due to degenerative spine diseases. Instances of work disability days increased for these patients from about 20 days three years before surgery to over 70 days in the year after surgery and declined to about 30 days in the third year following surgery. Work disability days caused by other diagnoses remained stable at a low level in this group. In LBP-patients without decompression surgery, work disability days were also mainly attributed to degenerative spine diseases and, to a lesser extent, to other somatic diagnoses. Again, there was a peak in this group in the first year following CED due to degenerative spine diseases, while work disability days for other diagnoses remained stable. Notably, the LBP-patients who did not receive decompression surgery had a higher level of work disability due to mental diagnoses throughout the observation years than those with LSDS. In the group with no diagnosed LBP, trends in different diagnosis-specific work disability days remained stable on a low level, however, most commonly caused by somatic diagnoses other than musculoskeletal, followed by mental diagnoses other than CMDs, degenerative spine diseases, and CMDs.

Table 3 shows the relative risks of slopes in diagnosis-specific work disability days among individuals in the three groups of interest. This table shows that, generally, there was an upward slope before CED, and a downward slope in the years after inclusion. This pattern was especially pronounced in patients with LBP who had work disability days due to degenerative spine diseases, and even more so in patients with LSDS. Concerning mental diagnoses, although a downward slope in work disability days due to CMDs was observed during the three years following CED, for other mental diagnoses an upward slope was observed during the same period.

Appendix A shows the frequencies and proportions of individuals with work disability (LTSA and DP) due to certain diagnoses in the three years following CED. In patients with LSDS, almost one fourth had LTSA, and more than 6% had DP in the three years following surgery. These proportions were higher than in LBP-patients without LSDS, and in individuals without LBP. In patients with LSDS, by far the most common diagnosis of LTSA and DP (46% of all LTSA and 59% of all DP) was musculoskeletal diseases, whereas 13% of LTSA and 17% of DP cases were due to mental diagnoses. On the other hand, mental diagnoses were the underlying cause for more than 22% of LTSA and 26% of DP in LBP-patients without LSDS during the three years after CED. Musculoskeletal diseases were again the most common reasons for LTSA and DP (50% of all LTSA and 46% of all DP) in LBP patient without LSDS. In contrast, the most common reason for LTSA and DP among individuals without diagnosed LBP was mental disorders (35% of all LTSA and 44% of all DP). Notably, the proportion of individuals being on LTSA or DP due to disorders of the nervous system, symptoms, signs and abnormal clinical laboratory findings, injuries/poisoning or other reasons was clearly higher in patients with LBP with or without LSDS, compared with individuals with no LBP.

## 4. Discussion

Our study showed trends of diagnosis-specific work disability in patients with lumber spine decompression surgery, as well as in LBP-patients without surgical treatment and individuals with no diagnosed LBP, during the three years before and after the cohort entry date. LBP-patients treated by LSDS had high amount of work disability due to degenerative spine diseases throughout the observation period, with a sharp downward trend during the second year following surgery. However, work disability due to degenerative spine diseases was still about 25% higher three years after than surgery than it was three years before surgery. In addition to musculoskeletal diseases, patients with LSDS also had high work disability caused by other somatic and mental diagnoses, before and after the surgery, indicating for many mental and somatic co-morbidities that are not affected by the improvement in degenerative lumber spine disease. Among all three groups, LBP-patients without LSDS had the highest proportion of patients and mean number of days with work disability due to somatic diagnoses other than musculoskeletal, CMDs, or other mental diagnoses throughout the study. 

The findings show that patients with LSDS, have a relatively poor long-term prognosis in terms of work disability due to degenerative spine diseases, although it seems that the surgery was able to curb the increasing work disability trend from the second postoperative year. In these patients, about one sixth had long term SA or DP due to musculoskeletal diseases in the three years following decompression surgery (which was only about 10% in LBP-patients without LSDS, and only about 2% in individuals without LBP). Work disability in patients with LSDS was already high at the beginning of the study and was even higher three years following the decompression surgery than three years before the intervention. According to recommendations, LSDS is performed as the primary option of therapy in severe cases of degenerative diseases, such as hernias and stenosis, and in rare acute cases of severe neurological complications [6,24], and more often electively in patients whose symptoms do not sufficiently respond to other medical and interventional treatment options [6,25]. Thus, the patients who undergo LSDS are generally those with a heavier disease burden than patients with LBP who are not treated with decompression surgery. Our results also suggest that patients with LSDS used more often and higher dosages of analgesic and other drugs than LBP-patients with no LSDS and individuals not diagnosed with LBP. This may be a probable explanation for why patients with LSDS have a poor prognosis in terms of work disability. Additionally, about 10% to 24% with LSDS develop severe side effects after surgery, including infections, hematoma, nerve root injury and the risk of re-operation, or general complications, like coronary ischemia, respiratory distress, and stroke [7]. This might further contribute to a high amount of work disability and the fact that work disability three years after surgery was even higher than three years before surgery. 

LBP-patients without decompression surgery had nearly double the amount of mean annual days of work disability due to CMDs and other somatic disorders—rather than musculoskeletal disorders—than the individuals without LBP. Similarly, the prevalence of work disability days due to degenerative spine diseases was three times higher, due to other musculoskeletal diseases was four times higher and due to mental diagnoses other than CMDs was slightly higher among LBP-patients without LSDS than in individuals without LBP. On the other hand, compared to individuals without LBP, LBP-patients with decompression surgery had similar amounts of mean annual work disability days for CMDs and other mental disorders, but six times higher rates of work disability days for degenerative spine diseases, three times higher for other musculoskeletal diseases, and one and a half times higher for other somatic disorders. Interestingly, work disability patterns due to diagnoses other than musculoskeletal diseases remained flat throughout the six observation years and did not show any upwards or downwards slope before or after CED, as it did for work disability due to degenerative spine diseases. Thus, patients with LBP (including those with LSDS) represent a population with many mental and somatic co-morbidities that were already present three years before CED (date of diagnosis/surgery) and did not change much over time. In particular, mental co-morbidities (particularly CMDs, i.e., depressive and anxiety disorders) have been shown to significantly increase work disability when co-occurring with chronic pain [2,10] or triggering other negative clinical outcomes following lumbar spine surgery, ranging from low quality of life to higher mortality [26]. Moreover, work disability and CMDs have also been shown to be associated with an adverse psychosocial work environment, such as job strain, job insecurity, bullying and effort-reward imbalance [27,28]. In fact, mental diagnoses are, besides musculoskeletal diseases, the most common reason for work disability in patients who initially had SA due to LBP. Other common diagnoses following work disability in those patients include injuries and poisoning, respiratory diseases, infectious diseases, and diseases of the digestive system [13]. In order to obtain a comprehensive picture of work disability in patients with low back pain, work disability days due to musculoskeletal illness and work disability days due to something other than musculoskeletal illness have to be considered in parallel.

An unexpected result was that work disability due to CMDs or due to other mental disorders was higher in individuals with LBP with no LSDS than it was in patients who had undergone LSDS. A possible explanation could be that the patients who already underwent LSDS and were granted SA or DP anytime during the three years after the surgery were, by default, recorded as SA or DP due to a degenerative spine disease that led to the surgery, although the underlying cause for work disability might have been mental or somatic co-morbidities. Another justification could be the obvious morphological defect and/or functional deficit in patients with LSDS [7], which eventually contributed more to work disability than the comorbid mental diagnoses. In contrast, in patients with LBP, in general, the bio-psycho-social concept (where psychological and social factors play an equal role in the development of adverse outcomes as biological factors) is well-established [29], and, therefore, in LBP-patients with no surgical treatment mental diagnoses might be indicated in SA and DP spells more frequently.

### Strengths and Limitations

This study is based on data from Swedish population-based nationwide registers, which have proofed to be of high quality [16,17,18,19,20,30]. Another strength was the use of a large study population (i.e., all individuals aged 19–60 with LSDS during the studied period in Sweden) and the two matched comparison groups. This method eliminated the possibility of selection and recall bias regarding exposure and outcome measures and avoided the loss to follow-up. Furthermore, a wide range of possible confounders, such as socio-demographic and clinical factors, could be included in the analyses. Additionally, in this study, work disability was used as the outcome parameter, which is only seldom used in longitudinal studies with patients undergoing LSDS. Furthermore, unlike other studies in the area where the usual approach is to consider work disability, in this study we looked at diagnosis-specific work disability leading to more specific and robust findings. Some limitations have to be mentioned. Patients with LBP were defined as having an LBP diagnosis at in- or specialised outpatient care. However, they might have had such diagnosis from primary care or have had LBP but did not seek help. Additionally, the national patient register does not provide information on the aetiology and pathology of the spinal lesion that led to LSDS, the indication for surgery, and disease severity. Furthermore, shorter SA spells (<14 days) could not be included in this study, which may lead to an underestimation of work disability.

## 5. Ethical Considerations

The study has been approved by the Regional Ethical Committee of Stockholm and by the ethical committees of Statistics Sweden, the National Board of Health and Welfare, and the Social Insurance Agency. After legal review, the data are only made available for researchers in the group who meet the criteria for access to this type of sensitive data. These procedures follow the Swedish Ethical Review Act, the Personal Data Act, and the Administrative Procedure Act. Data are derived from different registers. Therefore, no personal contact with the individuals is established. The integrity of the persons is secured through de-identification of the individual information by the data providing agencies. Results are presented on group level without any possibility of backward identification, as in numerous previous publications derived from this database. The research group applies high standards of data safety.

Key points:
This is the first register-based prospective cohort study on work disability following lumber decompression surgery, covering all individuals living in Sweden aged between 19–60 years.LBP-patients undergoing lumber spine decompression surgery (LSDS) had higher work disability during three years following surgery than patients without surgery and individuals without diagnosed LBP.LBP-patients undergoing LSDS had higher work disability three years following surgery than three years before, indicating unfavourable work disability prognosis, due to degenerative spine diseases.LSDS seems to be able to restrict the increasing work disability trend due to degenerative spine disease in the LBP-patients in the long run.Interventions and disability policy should focus on LBP-patients with degenerative spine disease.

## 6. Conclusions

This study showed that patients undergoing lumber spine decompression surgery have an unfavourable work disability prognosis, with already high work disability being primarily due to dorsopathies before surgery. Work disability in patients with LSDS due to other diagnoses is also high, but without clear trends over time from before to after surgery, indicating high mental and somatic co-morbidities. Despite the higher work disability three years following the decompression surgery than three years before, the trend shows that the decompression surgery seemed to restrict a further incline in work disability in the long run.

## Figures and Tables

**Figure 1 ijerph-18-08937-f001:**
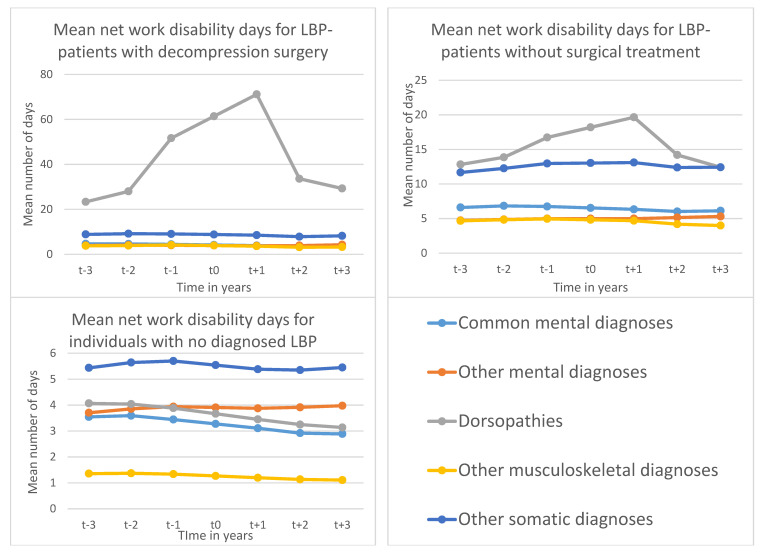
Mean number of diagnosis-specific ^1^ work disability days ^2^ among patients aged 19–60 years, living in Sweden, with low back pain (LBP) treated by lumber spine decompression surgery (LSDS) during 2008–2010, with LBP without LSDS and individuals with no diagnosed LBP. ^1^ Common mental diagnoses: ICD-10: F32-33, F40-43; Other mental diagnoses: ICD 10: F00-99, except F32-33, F40-43; Dorsopathies = Degenerative spine diseases: ICD-10: M40-54; Other musculoskeletal diagnoses: ICD 10: M00-99 except M40-54; Other somatic diagnoses: ICD 10: all somatic diagnoses except musculoskeletal diagnoses, uncomplicated delivery (O80), external causes of morbidity (V01-X59), factors influencing health status and contact with health services (Z00-Z99) and codes for special purposes (U00-U85) ^2^ Adjusted for age, sex, area of living, family situation, region of birth, level of education at cohort entry.

**Table 1 ijerph-18-08937-t001:** Socio-demographic characteristics at cohort entry of the patients aged 19–60, living in Sweden, with Lumber Spine Decompression Surgery (LSDS) during 2008–2010 in Sweden.

**Socio-Demographic Factors**	N (%) =10,800 (100%)
**Sex**	
Women	5100 (52.8)
Men	5700 (47.2)
**Age (in years)**	
19–24	376 (3.5)
25–34	1339 (12.4)
35–44	2662 (24.7)
45–54	2623 (24.3)
55–60	3800 (35.2)
**Education (in years)**	
Low (<10)	2161 (20.0)
Medium (10–12)	5632 (52.2)
High (>12)	3007 (27.8)
**Type of living area ^1^**	
Big cities	4107 (38.0)
Medium-sized cities	3688 (34.2)
Small cities/villages	3005 (27.8)
**Region of birth**	
Sweden	9127 (84.5)
Nordic countries (except Sweden)	503 (4.7)
EU-25 Europe (except Nordic countries)	226 (2.1)
Rest of the world	944 (8.7)
**Family situation**	
Married/cohabiting without children living at home	2424 (22.4)
Married/cohabiting with children living at home	3332 (30.9)
Single ^2^ without children living at home	3983 (36.9)
Single with children living at home	905 (8.4)
Aged ≤ 20 years living with parents	156 (1.4)

^1^ Type of living area: Big cities: Stockholm, Gothenburg and Malmo; Medium-sized cities: cities with more than 90,000 inhabitants within 30 km distance from the centre of the city; small cities/villages. ^2^ Single includes divorced, separated, or widowed.

**Table 2 ijerph-18-08937-t002:** Clinical factors three years before cohort entry date among Swedish residents, aged 19–60 years, with low back pain (LBP) treated by lumber spine decompression surgery (LSDS) during 2008–2010, with LBP without LSDS and persons with no diagnosed LBP.

	LSDS *n* (%) 10,800 (100%)	LBP No LSDS *n* (%) 109,179 (100%)	No Diagnosed LBP *n* (%) 472,191 (100%)
**Previous in- or specialised outpatient care due to**			
Common mental disorders ^1^*	644 (6.0)	8981 (8.2)	17,492 (3.7)
Other mental disorders ^2^*	974 (9.0)	13,309 (12.2)	29,579 (6.3)
LBP ^3^*	9725 (90.1)	15,342 (14.1)	-
Degenerative spine diseases other than LBP ^4^*	922 (8.5)	6453 (5.9)	-
Somatic diagnoses other than musculoskeletal diseases ^5^*	7453 (69.0)	79,533 (72.8)	241,467 (51.1)
**Sickness absence in the year prior to inclusion ***			
1–90 days	2791 (25.8)	17,084 (15.7)	32,453 (6.9)
91–180 days	1148 (10.6)	5175 (4.7)	5792 (1.2)
181–365 days	992 (9.2)	5017 (4.6)	4887 (1.0)
Initiated in the year prior to inclusion	373 (3.5)	2069 (1.9)	1631 (0.4)
**Analgesics during the year prior to inclusion (N02A and N02B in DDDs) ***			
Low (>0 to 0.5)	5909 (54.7)	38,811 (35.6)	46,280 (9.8)
Medium (>0.5 to 1.5)	2073 (19.2)	2214 (8.1)	6106 (1.3)
High (>1.5)	605 (5.6)	3672 (3.4)	1725 (0.4)
**Anxiolytics during the year prior to inclusion (N05B in DDDs) ***			
Low (>0 to 0.5)	1315 (12.2)	10,444 (9.6)	19,355 (4.1)
Medium (>0.5 to 1.5)	147 (1.4)	1624 (1.5)	2734 (0.6)
High (>1.5)	75 (0.7)	943 (0.9)	1336 (0.3)
**Antidepressants during the year prior to inclusion (N06A in DDDs) ***			
Low (>0 to 0.5)	920 (8.5)	8511 (7.8)	13,829 (2.9)
Medium (>0.5 to 1.5)	826 (7.7)	7599 (7.0)	19,651 (4.2)
High (>1.5)	439 (4.1)	3856 (3.5)	8425 (1.8)
**Sedative/hypnotics during the year prior to inclusion (N05C in DDDs) ***			
Low (>0 to 0.5)	1024 (9.5)	9812 (9.0)	20,471 (4.3)
Medium (>0.5 to 1.5)	564 (5.2)	5003 (4.6)	8946 (1.9)
High (>1.5)	299 (2.8)	2482 (2.3)	3603 (0.8)

* Significant between group differences tested by Chi2 at level of *p* < 0.001. ^1^ CMDs, International Classification of Diseases version 10 (ICD-10): F32-33, F40-43. ^2^ Other mental diagnoses, ICD-10: F00-99, except for F32-33, F40-43. ^3^ LBP, ICD-10: M47.8, M47.9, M47.9K, M48.0, M48.0K, M48.8, M48.8K, M48.8W, M51.1, M51.1K, M53.8, M53.9, M54.1, M54.3, M54.4 and M54.5. ^4^ Degenerative spine diseases other than LBP, ICD-10: M40-54 except codes for LBP (as mentioned above). ^5^ Somatic diagnoses other than musculoskeletal diseases, ICD 10: all somatic diagnoses except musculoskeletal diseases, uncomplicated delivery (O80), external causes of morbidity (V01-X59), factors influencing health status and contact with health services (Z00-Z99) and codes for special purposes (U00-U85).

**Table 3 ijerph-18-08937-t003:** Relative risk (RR) with confidence intervals (CI) of trends in diagnosis-specific work disability net days ^1^ among Swedish residents, aged 19–60 years, with low back pain (LBP) treated by lumber spine decompression surgery (LSDS) during 2008–2010, with LBP without LSDS and individuals with no diagnosed LBP.

Outcome Measures in Relation to Diagnosis-Specific Work Disability Days	Pre-Inclusion Period (Year t − 1 vs. t − 3 ^^^) RR (95% CI)	Near Inclusion Period (Year t + 1 vs. t − 1 ^^^) RR (95% CI)	Post-Inclusion Period (Year t + 3 vs. t + 1 ^^^) RR (95% CI)
**Work disability days due to degenerative spine diseases**			
LSDS	2.21 (2.12–2.32)	1.38 (1.35–1.41)	0.41 (0.39–0.43)
LBP no LSDS	1.30 (1.29–1.32)	1.18 (1.16–1.19)	0.63 (0.62–0.64)
No diagnosed LBP	0.91 (0.90–0.92)	0.89 (0.88–0.90)	0.86 (0.82–0.91)
**Work disability days due to other musculoskeletal diseases**			
LSDS	1.10 (1.05–1.16)	0.87 (0.83–0.90)	0.89 (0.83–0.96)
LBP no LSDS	1.06 (1.04–1.08)	0.95 (0.94–0.96)	0.85 (0.83–0.87)
No diagnosed LBP	0.99 (0.98–1.00)	0.90 (0.89–0.90)	0.93 (0.91–0.94)
**Work disability days due to CMDs**			
LSDS	0.95 (0.89–1.03)	0.88 (0.83–0.93)	1.00 (0.90–1.11)
LBP no LSDS	1.02 (1.00–1.04)	0.94 (0.92–0.96)	0.97 (0.94–0.99)
No diagnosed LBP	0.97 (0.96–0.98)	0.90 (0.89–0.91)	0.93 (0.92–0.94)
**Work disability days due to other mental disorders**			
LSDS	1.03 (0.95–1.10)	0.94 (0.89–0.99)	1.13 (1.02–1.25)
LBP no LSDS	1.06 (1.04–1.08)	1.00 (0.99–1.02)	1.07 (1.04–1.09)
No diagnosed LBP	1.06 (1.06–1.07)	0.98 (0.98–0.99)	1.03 (1.02–1.04)
**Work disability days due to other somatic diagnoses**			
LSDS	1.02 (0.98–1.07)	0.94 (0.91–0.98)	0.96 (0.90–1.03)
LBP no LSDS	1.11 (1.10–1.13)	1.01 (1.00–1.02)	0.95 (0.93–0.96)
No diagnosed LBP	1.05 (1.04–1.06)	0.95 (0.94–0.95)	1.01 (1.00–1.02)

^1^ Adjusted for age, sex, area of living, family situation, region of birth, level of education. ^^^ Reference year.

## Data Availability

Due to the sensitive nature of the data, the data providers have restricted the public availability.

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
