# Peer review of "Diagnosis-Specific Work Disability before and after Lumbar Spine Decompression Surgery—A Register Study from Sweden"

_ijerph, 2021, doi:10.3390/ijerph18178937_

Round 1

Reviewer 1 Report

The paper is well composed and written. The topic is of interest for medical doctors who treat patients with low back pain. The study is aimed at the interactions between the pattern of diagnosis-specific work disability in patients before and after lumber spine decompression surgery with some socio-demographic and clinical factors. The study is confined by the individuals registered in Sweden but it is quite a representative group for statistical analyses. 

Author Response

We thank the reviewer for the positive feedback.

Reviewer 2 Report

  1. The analyzed results looks like good. But I wonder whether the distribution of data is proper or not. Some max values can drive to these results although you showed the results of CI.
  2. Does the increase of work disability due to CMDs mean the fear of workers because related companies do not consider the health of workers? More detail description will be helpful for the interest of readers, I think.
  3. For the cross-check or extension, I recommend you should open your data set, if possible. 

Author Response

Query 1. The analyzed results looks like good. But I wonder whether the distribution of data is proper or not. Some max values can drive to these results although you showed the results of CI.

Authors’ response: Thank you for your positive evaluation of our study and your important question regarding the distribution of the outcome and covariate measures and their effect on the results. Work disability was measured as the mean of the annual sum of net days with sickness absence or disability pension due to a specific diagnosis. This means that e.g. a person being 100% sickness absent during the entire year, has 365 work disability days. This further implies that the data were per definition of the outcome “truncated” at a certain level, as the sum of the work disability days can never exceed 365 or 366 in a leap year. The data also underwent considerable data cleaning taking care of e.g. overlapping spells. Consequently, it is impossible that the findings are driven by “unnatural” max values or outliers with exceptionally high values.

Moreover, we show confidence intervals, as you pointed out. The confidence intervals for the estimates reported in the manuscript are very narrow which is related to the exceptionally large size of the study population and suggest a good precision.

Query 2. Does the increase of work disability due to CMDs mean the fear of workers because related companies do not consider the health of workers? More detail description will be helpful for the interest of readers, I think.

Authors’ response: Indeed, work disability due to CMDs has been linked to an adverse psycho-social work environment. We have therefore added the following sentence with relevant references in the discussion section on page 9: “Moreover, work disability and CMDs have also been shown to be associated with an adverse psycho-social work environment, such as high job strain, job insecurity, bullying and effort-reward imbalance 27, 28”.

27 Niedhammer I, Sultan-Taïeb H, Parent-Thirion A, Chastang JF. Update of the fractions of cardiovascular diseases and mental disorders attributable to psychosocial work factors in Europe. Int Arch Occup Environ Health. 2021 Jun 28:1–15. doi: 10.1007/s00420-021-01737-4. Epub ahead of print. PMID: 34181059; PMCID: PMC8237556.

28 Helgesson M, Marklund S, Gustafsson K, Aronsson G, Leineweber C. Interaction Effects of Physical and Psychosocial Working Conditions on Risk for Sickness Absence: A Prospective Study of Nurses and Care Assistants in Sweden. Int J Environ Res Public Health. 2020 Oct 12;17(20):7427. doi: 10.3390/ijerph17207427. PMID: 33053900; PMCID: PMC7601317.

Query 3. For the cross-check or extension, I recommend you should open your data set, if possible.

Authors’ response: Unfortunately, the data cannot be made publicly available. According to the Swedish Ethical Review Act, the Personal Data Act, and the Administrative Procedure Act, data can only be made available, after legal review, for researchers who meet the criteria for access to this type of sensitive and confidential data.